# Novel Temperature/Reduction Dual-Stimulus Responsive Triblock Copolymer [P(MEO_2_MA-*co*- OEGMA)-*b*-PLLA-SS-PLLA-*b*-P(MEO_2_MA-*co*-OEGMA)] via a Combination of ROP and ATRP: Synthesis, Characterization and Application of Self-Assembled Micelles

**DOI:** 10.3390/polym12112482

**Published:** 2020-10-26

**Authors:** Fei Song, Zhidan Wang, Wenli Gao, Yu Fu, Qingrong Wu, Shouxin Liu

**Affiliations:** Key Laboratory of Applied Surface and Colloid Chemistry, Ministry of Education, School of Chemistry and Chemical Engineering, Shaanxi Normal University, Xi’an 710119, China; 17563713257@163.com (F.S.); 18189608643@163.com (Z.W.); 17853465526@163.com (W.G.); fy2247127430@163.com (Y.F.); wuqingrong123456@163.com (Q.W.)

**Keywords:** temperature/reductive dual-stimulus responsiveness, block copolymers, amphiphilic, ring-opening polymerization, atom transfer radical polymerization

## Abstract

Novel temperature/reduction dual stimulus-responsive triblock copolymers, poly [2-(2-methoxyethoxy) ethyl methacrylate-*co*-oligo (ethylene glycol) methacrylate]-*b*-(L-polylactic acid)-SS-*b*-(L-polylactic acid)-*b*-poly[2-(2-methoxyethoxy) ethyl methacrylate-*co*-oligo(ethylene glycol)methacrylate] [P(MEO_2_MA-*co*-OEGMA)-*b*-PLLA-SS-PLLA-*b*-P(MEO_2_MA-*co*-OEGMA)] (SPMO), were synthesized by ring opening polymerization (ROP) of L-lactide and 2,2’-dithio diethanol (SS-DOH), and random copolymerization of MEO_2_MA and OEGMA monomers via atom transfer radical polymerization (ATRP) technology. The chemical structures and compositions of the novel copolymers were demonstrated by proton nuclear magnetic resonance (^1^H NMR) and Fourier transform infrared spectroscopy (FTIR). The molecular weights of the novel copolymers were measured by size exclusive chromatography (SEC) and proved to have a relatively narrow molecular weight distribution coefficient (*Ð*M ≤ 1.50). The water solubility and transmittance of the novel copolymers were tested via visual observation and UV–Vis spectroscopy, which proved the SPMO had a good hydrophilicity and suitable low critical solution temperature (LCST). The critical micelle concentration (CMC) of the novel polymeric micelles were determined using surface tension method and fluorescent probe technology. The particle size and morphology of the novel polymeric micelles were characterized by dynamic light scattering (DLS) and transmission electron microscopy (TEM). The sol–gel transition behavior of the novel copolymers was studied via vial flip experiments. Finally, the hydrophobic anticancer drug doxorubicin (DOX) was used to study the in vitro release behavior of the novel drug-loaded micelles. The results show that the novel polymeric micelles are expected to become a favorable drug carrier. In addition, they exhibit reductive responsiveness to the small molecule reducing agent dithiothreitol (DTT) and temperature responsiveness with temperature changes.

## 1. Introduction

In recent years, amphiphilic copolymers have drawn much attention in the field of biomedicine due to their small size and loading capacity of hydrophobic drugs. For example, they can be used as the carrier for targeted drug delivery or the delivery tools for cancer chemotherapeutics. They also have great application prospects in gene therapy and diagnostic imaging [1,2,3,4]. The advantage of the copolymers is that they can self-assemble to form micelles of different shapes with a hydrophilic shell and a hydrophobic inner cavity in aqueous solution, such as sphere, hollow tubes, worm-like rods or large bubbles [5,6]. The polymeric micelles with specific structures provide internal space for guest molecules (such as hydrophobic drug molecules), thereby increasing the apparent solubility of anticancer drugs [7,8]. In order to regulate the release of encapsulated drug molecules, the designed high molecular copolymers can make corresponding changes to external stimuli (e.g., temperature, pH, light, ionic strength, redox, etc.) [9,10,11]. In the past few decades, single-stimulus-responsive copolymers have been widely developed, and multi-stimulus-responsive copolymers molecules also are gradually designed because of their responding to each stimulus environment accurately and independently. More importantly, they can co-regulate the drug release behavior under multiple stimuli [12]. Therefore, it is very meaningful to design a unique copolymer molecule with multiple stimulus responsiveness.

In the study of multiple-stimulus-responsive copolymers, Wu’s team designed a light- responsive amphiphilic random copolymer of polybornene containing functional spiropyran groups [13]. The copolymers can form spherical micelles with core–shell structure in water self-assembly, and the reversible encapsulation and release behavior of drug-loaded micelles can be realized under light stimulation. Lei’s team synthesized an inner-layer crosslinked tetrablock copolymer with temperature, pH, and reduction triple-stimuli-response as nanocarriers for drug delivery and release, which could self-assemble into non-crosslinked lamellar micelles in acidic aqueous solution to load the drug doxorubicin (DOX) [14]. Among all stimulus responses, temperature stimulation has unique advantages and can be easily applied to research [15,16]. It is well know that the widely studied thermo-responsive hydrophilic copolymers include poly(*N*-vinylcaprolactam) (PNVCL) [17,18], poly (*N*-isopropylacrylamide) (PNIPAAm) [19,20], poly(2-(2-methoxy ethoxy) ethylmethacrylate) (PMEO_2_MA) and poly(oligo (ethylene glycol) methacrylate) (POEGMA) [21,22,23], and poly[tri(ethylene glycol) methyl ether methacrylate] (PM3) [24,25,26]. Oligomer ethylene glycol methyl ether methacrylate P(MEO_2_MA-*co*-OEGMA) has appealed to much attention because of its advantages of temperature sensitivity, water solubility, and biocompatibility [26,27]. In the literature report of the J. F. Lutz’s team [28], the low critical solution temperature (LCST) of PMEO_2_MA and POEGMA are 26 °C and 90 °C, respectively. In addition, the LCST value of random copolymer P(MEO_2_MA-*co*-OEGMA) is adjusted by controlling the amount of OEGMA. In addition, since the reductive glutathione (GSH) concentration is known to be 100–1000 fold higher in tumor cells than in the normal extracellular fluid, the disulfide bonds can be cleaved to form sulfhydryl groups by reducing agents. Therefore, the micelles crosslinked with disulfide linkages have been investigated as drug carries for tumor-specific delivery [29,30,31,32,33].

However, most anticancer drugs suffer from insolubility in water, such as platinum anticancer drugs, which can cause several undesirable side effects, including nephrotoxicity and neurotoxicity. These properties limit the future use of preparations [34,35,36]. In recent years, the development of amphiphilic copolymers has solved this difficulty. They can form core–shell micelles with self-assembly in water such that they achieve reversible coating and release behaviors for drug-loaded micelles under different stimuli. Recently, polylactic acid (PLA), a biodegradable hydrophobic copolymer, has been found to have excellent biocompatibility and bioabsorbability. Therefore, researchers are increasingly interested in researching amphiphilic copolymers with hydrophobic PLA chain [37,38]. It is generally known that lactic acid has three types of isomers, namely L-lactic acid (LLA), D-lactic acid (DLA), and D,L-lactic acid (DLLA). For the corresponding copolymer, PLLA and PDLA are semicrystalline copolymers with good mechanical strength, whereas PDLLA is an amorphous copolymer that is easy to decompose and has poor strength and durability. Therefore, PLLA and PDLA are the main hydrophobic segments materials for the synthesis of amphiphilic copolymers.

Therefore, in this research work, we design the temperature/reduction dual stimulus response triblock copolymer via ring-opening polymerization (ROP) [39,40] and atom transfer radical polymerization (ATRP) [41,42] technology, in which the hydrophobic segment is PLLA-SS-PLLA and the hydrophilic segment is P(MEO_2_MA-*co*-OEGMA). The study on the properties of the synthesized amphiphilic copolymer and the drug release behavior not only proves that the spherical micelles formed from self-assembly in water are a good carrier for hydrophobic anticancer drugs, but also the drug-loaded micelles have a precise targeting effect in tumor cells under the co-stimulation of temperature and reducing environment. It promotes the application of multiple stimulus-responsive amphiphilic polymeric micelles in the biomedical field.

## 2. Materials and Methods

### 2.1. Materials

Tin(II) 2-ethylhexanoate (Sn(Oct)_2_) was purchased from Bailingwei (Beijing, China). L-lactide (LLA) and oligo(ethylene oxide) methacrylate (OEGMA, 95%, *M*_n_ = 475 g·mol^−^^1^) were obtained from Aladdin (Shanghai Aladdin Biochemical Technology Co., Ltd., Shanghai, China). 2-(2-methoxyethoxy) ethyl methacrylate (MEO_2_MA, 95%, *M*_n_ = 188.22 g·mol^−1^) was purchased from TCI (Shanghai, China). 2,2’-dithio diethanol (SS-DOH) was obtained from Alfa Aesar (Shanghai, China). 2-Bromo-2-methylpropionyl bromide (Br-iBuBr) was obtained from Bailingwei (Beijing, China). 2,2′–bipyridine (bpy) was obtained from Guangdong Guanghua Technology Co., Ltd. (Guangdong, China). CuBr was synthesized using oxydoreduction of KBr, CuSO_4_ and Na_2_SO_3_. 1,4-Dithiothreitol (DTT) was purchased from Rhawn (Rhawn Co., Ltd., Shanghai, China). All solvents need to be distilled prior to use. Toluene was dried with sodium wire, triethylamine (TEA) was dried with CaH_2_, tetrahydrofuran (THF) was dried with sodium and copper sulfate, deionized water was used as an aqueous solution, and other reagents were obtained from commercial sources and can be used directly without further purification.

### 2.2. Synthesis

#### 2.2.1. Synthesis of Homopolymer OH-PLLA-SS-PLLA-OH by ROP

Synthesis of OH-PLLA-SS-PLLA-OH used the method of ROP with L-lactide (LLA) as the polymeric raw material, 2,2’-dithio diethanol (SS-DOH) as the initiator, and Sn(Oct)_2_ as the catalyst [43]. The specific synthesis route was as follows. L-lactide (2.41 g, 16 mmol) and dry toluene (10 mL) were added to a dry three-necked flask (50 mL). After L-lactide was dissolved completely, the initiator SS-DOH (0.1 mL, 0.8 mmol) and the catalyst Sn(Oct)_2_ (0.013 mL, 0.04 mmol) were added to the above mixture, respectively. Finally, the mixture was heated to 120 °C and stirred for 24 h under protection of the nitrogen atmosphere. The reaction was cooled to room temperature to terminate the polymerization. The obtained oily liquid was precipitated twice in cold anhydrous ether, and then dried in a vacuum drying cabinet at room temperature for 12 h to obtain the white powdery solid products. The synthetic route was shown in Scheme 1. 

#### 2.2.2. Synthesis of Macromolecular Initiator iBuBr-PLLA-SS-PLLA-iBuBr

Synthesis of macromolecular initiator iBuBr-PLLA-SS-PLLA-iBuBr was performed by esterification reaction with dry triethylamine (TEA) as the acid-binding reagent, and 2-bromo-2- methylpropionyl bromide (Br-iBuBr) and purified homopolymers OH-PLLA-SS-PLLA-OH as reactants [43,44]. The specific synthesis route was as follows. The purified, dried homopolymer OH-PLLA-SS-PLLA-OH (0.46 g, 0.15 mmol), dried tetrahydrofuran (25 mL), and triethylamine (0.4 mL, 2.9 mmol) were added to a single-necked round-bottom flask (50 mL). After homopolymers were dissolved completely, Br-iBuBr (0.36 mL, 2.9 mmol) was added dropwise for 30 min in an ice bath at 0 °C and then kept at room temperature for 24 h. The formed solids (HCl:Et_3_N adducts) were removed by filtration and then obtaining the yellow oily crude products by rotary evaporation. The crude products were purified from precipitation with cold methanol to remove excess TEA and Br-iBuBr, which were soluble in MeOH. Finally, the precipitates were collected by vacuum filtration, and dried in a vacuum drying oven at room temperature for 12 h to obtain the white solid products. The synthetic route was shown in Scheme 1.

#### 2.2.3. Synthesis of Triblock Copolymer [P(MEO_2_MA-co-OEGMA)-b-PLLA-SS-PLLA-b-P(MEO_2_MA- co-OEGMA)] by ATRP

The temperature/reduction dual stimulus responsive triblock copolymer were synthesized by ATRP technology with iBuBr-PLLA-SS-PLLA-iBuBr as the macromolecular initiator and 2 -(2-methoxyethoxy)ethyl methacrylate (MEO_2_MA) and oligo(ethylene glycol) methacrylate (OEGMA) as the monomers for polymerization. The specific synthesis route was as follows. The macroinitiator iBuBr-PLLA-SS-PLLA-iBuBr (0.173 g, 0.052 mmol), *N*,*N*-dimethylformamide (2 mL), MEO_2_MA (2.4 mL, 13 mmol), and OEGMA (0.3 mL, 0.68 mmol) were added to a dried Schlenk flask (50 mL). After the mixture dissolved completely, the reaction system was deoxygenated by four freeze–pump–thaw cycles. When the reaction flask was filled with nitrogen again, the ligand 2,2-bipyridine (0.016 g, 0.104 mmol) and the catalyst CuBr (0.008 g, 0.052 mmol) were added into the mixture solution. After the reaction system was sealed, the mixture solution was heated to 70 °C and stirred for 12 h under protection of the nitrogen atmosphere. The reaction was cooled to room temperature to terminate the polymerization. The mixture was precipitated with cold anhydrous ether to obtain light green viscous substances. The copolymers diluted with deionized water were transferred to an MWCO14kDa dialysis bags for four days, and then freeze-dried for getting the colorless transparent gel-like products. The synthetic route was shown in Scheme 1. 

### 2.3. Characterization

#### 2.3.1. Nuclear Magnetic Resonance Spectroscopy (^1^H NMR)

The chemical structures and compositions of the novel stimulus-responsive triblock copolymers were determined by ^1^H NMR with CDCl_3_ as the solvent and tetramethylsilane (TMS) as the internal standard at room temperature using the Bruker 300 MHz spectrometer (300 MHz AVANCE, Bruker Corporation, Karlsruhe, Germany).

#### 2.3.2. Fourier-Transform Infrared Spectroscopy (FTIR)

The functional groups of the copolymers were detected by FTIR using the Tensor 27 infrared spectrometer (Bruker Corporation, Karlsruhe, Germany). Before being measured, the sample was uniformly mixed with dried KBr, and then dried in a vacuum drying oven for 6 h. Finally, the sample was compressed and tested.

#### 2.3.3. Size Exclusive Chromatography (SEC)

The number average molecular weight (*M*_n_) and molecular weight distribution coefficient (*Ð*M = *M*_w_/*M*_n_) of the polymers were measured by SEC using VISCOTEK TM size exclusive chromatography (Malvin Company, Malvin, UK) with monodisperse polystyrene (PS) as a standard sample to calibrate the column and chromatographically pure THF as the mobile phase, where the flow rate was set to 100 μL/min. The samples measured were filtered three times with a 0.22 μm organic filter (*Φ* 13 mm), and then put into the sample bottle to be tested.

### 2.4. The Properties of Copolymers

#### 2.4.1. Water Solubility and Temperature Sensitivity of Copolymers

The water solubility and temperature sensitivity were monitored by digital photos of the copolymeric aqueous solutions (1.00 mg/mL) in a transparent sample vial at 25 °C and 35 °C, respectively.

#### 2.4.2. Low Critical Solution Temperature (LCST) of Copolymers

The LCST of the novel copolymers and the relationship between the LCST and the concentration of copolymers were measured using UV–Vis spectroscopy (TU-1901, Beijing Purkinje General Instrument Corporation, Beijing, China). Firstly, the LCST of novel copolymers with different polymerization degrees (DP) were measured. A series of copolymer aqueous solutions (1.00 mg/mL) was arranged in a transparent sample vial. The transmittance of the copolymer aqueous solutions was measured using the TU-1901 in the temperature range of 20–70 °C. The data graph of temperature and transmittance was drawn via Origin8 (OriginLab, Waltham Mass, MA, USA) in order to get the LCST of the different DP of copolymers. Secondly, the relationship between the LCST and concentration of triblock copolymer aqueous solutions was explored. Different concentrations of copolymer aqueous solutions were configured, and then the transmittance was measured using the TU-1901 to obtain the relationship between LCST and concentration of copolymers aqueous solutions.

### 2.5. Properties of Copolymeric Micelles 

#### 2.5.1. Critical Micelle Concentration (CMC) of the Polymeric Micelles 

The CMC of the novel polymeric micelles were measured through two methods. The first way was to determine the CMC using fluorescent probe technology. Different concentrations of the polymeric micelle aqueous solutions were configured, and the concentration of pyrene was 1.0×10^−5^ mol/L in these solutions. They were measured using the PELS55 fluorescence spectrophotometer (PE Company, Waltham Mass, United States); wavelength range was set to 350–500 nm, and then fluorescence intensity *I*_1_ and *I*_3_ were recorded at 373 nm and 384 nm. Finally, taking *I*_3_/*I*_1_ as the ordinate and the concentration of the micellar solution as the abscissa to plot, the CMC was determined by extrapolation. The second way was to determine the CMC using the surface tension method. Different concentrations of the polymeric micelle aqueous solution were configured, and then were measured by the surface tension method. Finally, taking the surface tension as the ordinate and the concentration of the micellar solution as the abscissa to plot, the CMC was determined by the turning point. 

#### 2.5.2. Dynamic Light Scattering (DLS) of the Polymeric Micelles 

Dynamic light scattering of the novel polymeric micelles was determined using the particle size analyzer (BI 90Plus Laser Particle Analyzer, Brookhaven, New York, United States); the excitation wavelength and the scattering angle were set to 659.0 nm and 90.00°, respectively. A series of the polymeric micelle aqueous solutions (1.00 mg/mL) were arranged in the transparent sample vial. The samples measured were filtered with the 0.45 μm organic filter and then put into the sample bottle to be tested. The particle sizes of the copolymer micellar solution were measured severally at different ratios or external stimuli (such as temperature, reducing agent DTT).

#### 2.5.3. Morphology of the Polymeric Micelles

The morphology and size of the polymeric micelles were observed using JEOL JEM-2100 instrument (Tokyo, Japan) with an accelerating voltage 200 kV at different external stimuli (such as temperature, reducing agent DTT). A series of the polymeric micelle aqueous solution (1.00 mg/mL) were arranged in transparent sample vial. After equilibrating for 30 min under different external stimulus environments, drawing 10 μL of micellar solution and dropping it on a 300 mesh–carbon–support film. Then, the samples were dyed with 1 wt% phosphotungstic acid for 3 min. After they were dried naturally, the morphology and size of micelles were observed with transmission electron microscope under different stimuli.

### 2.6. Sol–Gel Transition of the Copolymers

The vial flip experiment was used to study the sol–gel transition behavior of copolymers. Different mass concentrations of the copolymeric aqueous solutions were configured in transparent sample vial. ZD-85 constant temperature steam bath oscillator (Guohua Enterprise, Suzhou, China) was used to set the temperature range of 25–55 °C, and the temperature interval was 5 °C. After each copolymeric aqueous solution was equilibrated at each temperature for 10 min, the state of the copolymers was recorded through digital photos to find the critical temperature of gelation of the copolymers.

### 2.7. In Vitro Release Behavior of Polymeric Drug-Loaded Micelles

#### 2.7.1. Preparation of Polymeric Micelles Loaded with Doxorubicin (DOX)

The hydrophobic anti-cancer drug (DOX) was loaded into the hydrophobic core of the micelles by the method of dialysis. Because the medicine DOX·HCl is a hydrophilic anticancer drug, it is necessary to convert DOX·HCl into a hydrophobic drug in order to research the in vitro release behavior of micelles. Therefore, this is why triethylamine is added when preparing drug-loaded micelles. Because triethylamine is a kind of acid-binding reagent, it can remove HCl of DOX·HCl to form HCl:Et_3_N adducts. The DOX without HCl is the hydrophobic drug and encapsulated in the micellar core easily. The specific prepared route was as follows. The amphiphilic triblock copolymers (50 mg), solvent DMF (4 mL), DOX·HCl (4 mg), and same molar amount of triethylamine were added in transparent sample vial. When the copolymers were dissolved completely, the mixtures were dropped into deionized water (20 mL) slowly in the ice bath. Finally, the mixtures were transferred to the MWCO14kDa dialysis bags for 48 h to remove DMF and free DOX, and then freeze-dried for getting the polymeric drug-loaded micelles.

#### 2.7.2. Accumulative Drug Release of Polymeric Drug-Loaded Micelles

The accumulative release of DOX from drug-loaded micelles was measured using the UV–Vis spectroscopy. First of all, different concentrations of DOX solutions (concentration range 1–50 μg/mL) were configured with PBS (pH = 7.4) buffer solution, and then the standard curve of DOX-PBS was obtained by measuring the absorbance of the free DOX solution. Secondly, the drug-loaded micellar solutions (1 mg/mL) were configured with PBS buffer solution (pH = 7.4). The drug-loaded micelles were transferred to the MWCO3kDa dialysis bags, after they were dissolved completely. The study of drug-sustained release was carried out in PBS buffer solution (250 mL) under different external stimuli. Taking out the released solution (4 mL) within a certain time interval and measuring the absorbance at 482.5 nm by the ultraviolet spectrophotometer, adding fresh PBS buffer solution (4 mL) of the same kind to the drug release system synchronously. Finally, the accumulative release of DOX from drug-loaded micelles was calculated according to the standard curve and the following formula:

$$\mathrm{Accumulative}\;\mathrm{release}(\%)=\frac{V_{e}\sum_{1}^{n-1}c_{i}{+V}_{0}c_{n}}{m_{\mathrm{drug}}}\times 100$$
where *V*_e_ represents the volume of the release solution taken out each time, *V*_0_ represents the total volume of the release solution, *c*_i_ represents concentration of the sample withdrawn at the interval of *t*_i_, and *m*_drug_ represents the total mass of DOX-loaded polymeric micelles in the measurement system.

## 3. Results and Discussion

### 3.1. Synthesis and Characterization of Temperature/Reduction Dual-Stimulus Responsive Triblock Copolymers 

Temperature/reduction dual-stimulus responsive triblock copolymers were synthesized via ROP and ATRP. Firstly, synthesis of OH-PLLA-SS-PLLA-OH used ROP with LLA as the polymerization raw material, SS-DOH as the initiator, and Sn(Oct)_2_ as the catalyst. Secondly, synthesis of the macromolecular initiator iBuBr-PLLA-SS-PLLA-iBuBr by esterification reaction with dry TEA as the acid-binding reagent and Br-iBuBr and purified homopolymers OH-PLLA-SS-PLLA-OH as reactants. Finally, temperature/reduction dual-stimulus responsive triblock copolymers were synthesized with different DP by ATRP technology with iBuBr-PLLA-SS-PLLA-iBuBr as the macromolecular initiator and MEO_2_MA and OEGMA as the monomers for polymerization. Regarding the adjustment in the polymerization degree, the human body normal temperature was taken as the optimal ratio, and then the same intervals were expanded or reduced to synthesize the copolymers of the control ratios on this basis. The polymerization information of the block copolymers is shown in Table 1.

#### 3.1.1. ^1^H NMR and FTIR Temperature/Reduction Dual-Stimulus Responsive Triblock Copolymers

The chemical structures and compositions of the novel triblock copolymers were determined by ^1^H NMR and FTIR. Figure 1 shows the FTIR spectrum of the SP and SPBr. In the FTIR spectrum of OH-PLLA-SS-PLLA-OH, the peaks at 1750 cm^−1^ and 3650 cm^−1^ are the stretching vibration absorption peak of C=O and -OH, respectively, the peaks at 1189 cm^−1^ is the stretching vibration absorption peak of C-O-C, the peaks at 2945 cm^−1^ is the stretching vibration absorption peak of C-H for methyl and methylene, and the peaks at 1647 cm^−1^ and 1378 cm^−1^ are the in-plane flexural vibration absorption peaks of CH- for methylene and methyl, respectively. The peak at 1378 cm^−1^ is a sign of the presence of methyl groups. These peaks indicate that the synthesis of homopolymers (OH-PLLA-SS-PLLA-OH) is successful. In the iBuBr-PLLA-SS-PLLA-iBuBr FTIR spectrum, it can be obviously seen that the stretching vibration absorption peak of -OH at 3650 cm^−1^ disappears compared to the FTIR spectrum of the OH-PLLA-SS-PLLA-OH. It shows that the terminal hydroxyl of the homopolymer has an esterification reaction with Br-iBuBr, and the macromolecular initiator is synthesized successfully.

Figure 2 shows the ^1^H NMR spectrum of the three-step products. In the ^1^H NMR spectrum of OH-PLLA-SS-PLLA-OH, the peak at 5.18 ppm (peak “a”) is attributed to the methine proton of PLLA repeat units, the peaks at 2.84 ppm (peak “c”), and 4.38 ppm (peak “b”) are attributed to the methylene proton peak of the -OCH_2_CH_2_-SS-CH_2_CH_2_O- group in 2, 2’-dithio diethanol, the peak at 1.59 ppm (peak “d”) is attributed to the methyl proton peak of PLLA repeat units. These proton peaks indicate that the synthesis of homopolymers is successful. In the ^1^H NMR spectrum of iBuBr-PLLA-SS-PLLA-iBuBr, it can be obviously seen that the methyl proton peak of 2,2’-dithio diethanol at 2.00 ppm (peak “e”) appears compared to the ^1^H NMR spectrum of the homopolymer. It shows that the terminal hydroxyl of the homopolymer has an esterification reaction with Br-iBuBr, and the macromolecular initiator is synthesized successfully. In the ^1^H NMR spectrum of the triblock copolymers P(MEO_2_MA-*co*-OEGMA)-*b*-PLLA-SS-PLLA-*b*-P(MEO_2_MA-*co*-OEGMA), some new proton peaks appeared in addition to the methyl and methine proton peaks of PLLA repeat units. The peaks at 4.00 ppm (peak “f”) and 3.60 ppm (peak “g”) are attributed to the methylene proton peak of repeating units adjacent to the ester bond of MEO_2_MA and OEGMA. The peaks at 3.40 ppm (peak “h”) is attributed to the terminal methyl proton peaks of MEO_2_MA and OEGMA. The peaks at 1.82 ppm (peak “i”) and 0.87 ppm~1.05 ppm (peak “j”) are attributed to the methylene and methyl proton peaks of MEO_2_MA and OEGMA on the main chain of the triblock copolymer, respectively. The existence of these peaks proves that the temperature-sensitive monomers MEO_2_MA and OEGMA have been successfully copolymerized at random. In conclusion, the novel stimulus-responsive triblock copolymers are synthesized successfully.

#### 3.1.2. Size Exclusive Chromatography (SEC) of Temperature/Reduction Dual-Stimulus Responsive Triblock Copolymers 

Table 1 shows that the number average molecular weight (*M*_n_) and molecular weight distribution coefficient (*Ð*M = *M*_w_/*M*_n_) of the novel stimulus-responsive block copolymers were measured using VISCOTEK TM gel permeation chromatography (Malvin Company, Malvin, UK). According to the information in Table 1, it can be known that the measured molecular weights are all basically the same as the theoretical molecular weights, and the molecular weight distribution coefficient (*Ð*M ≤ 1.50) is relatively narrow. Figure 3 shows that SEC traces of SPMO2 copolymer without and with 10 mM DTT. It can be obviously seen that *M*_n_ of the degraded copolymer decreases. Therefore, the triblock copolymers with the disulfide bond have been synthesized.

### 3.2. The Properties of Temperature/Reduction Dual-Stimulus Responsive Triblock Copolymers 

#### 3.2.1. Water Solubility and Temperature Sensitivity of Copolymers

Figure 4 shows the water solubility and temperature sensitivity of amphiphilic triblock copolymers. It is well known that PLLA are the biodegradable, nontoxic, and hydrophobic copolymers, and P(MEO_2_MA-*co*-OEGMA) are the hydrophilic random copolymers with temperature sensitivity. Therefore, P(MEO_2_MA-*co*-OEGMA)-*b*-PLLA-SS-PLLA-*b*-P(MEO_2_MA-*co*-OEGMA) is an amphiphilic dual stimulus-responsive triblock copolymer. The polymeric degree (DP) of the hydrophilic segment is much higher than that of the hydrophobic segment from the Table 1, so the copolymers exhibits obvious water solubility. As shown in Figure 4a, SPMO1–SPMO5 are all clear and transparent copolymeric aqueous solutions. Because the oligo (ethylene glycol) side chain in the hydrophilic segment forms hydrogen bonds with water molecules, it promotes the water solubility of the copolymers at room temperature. It can be known that the water solubility of the copolymers increases, with the polymerization degree of hydrophilic segments increases by comparing SPMO1, SPMO2, and SPMO3. It can be seen from Figure 4b that the copolymeric aqueous solutions become opaque and slightly blue when the temperature rises to 35 °C. We found that the higher the content of OEGMA, the higher the transition temperature of the copolymers by comparing SPMO2, SPMO4, and SPMO5. In summary, Figure 4 shows that the SPMO triblock copolymers have the favorable water solubility and temperature responsiveness.

#### 3.2.2. Low Critical Solution Temperature (LCST) of Copolymers

Figure 5 shows the transmittance curve of SPMO triblock copolymeric aqueous solutions with different proportions at different temperatures. Figure 5a is the transmittance curve of SPMO1–SPMO5 aqueous solutions (1.00 mg/mL) at different temperatures. It can be seen that the LCST of SPMO1, SPMO2, and SPMO3 are 33 °C, 37 °C, and 39 °C, respectively. This means that the higher the DP of the hydrophilic segments, the higher the LCST, and the longer DP of the hydrophilic segments, the stronger the hydrogen bond with water molecules. It takes a higher temperature to break the hydrogen bond, so the LCST will increase. The LCST of SPMO2, SPMO4, and SPMO5 are 37 °C, 31 °C, and 42 °C, respectively, from Figure 5a. Therefore, when the DP of the hydrophilic segments are equal, the higher the content of temperature-sensitive monomer OEGMA, the higher the LCST. According to literature reports, the LCST of POEGMA is 90 °C, and the LCST of PMEO_2_MA is 26 °C [45]. Therefore, when OEGMA and MEO_2_MA are copolymerized randomly, LCST will be close to the high-content temperature-sensitive copolymers. Figure 5b is the transmittance curve of SPMO2 copolymeric aqueous solutions with different concentrations at different temperatures. Figure 5c explores the changes of LCST for the copolymers under different concentrations. The LCST decreases as the copolymers concentration increases through the analysis of Figure 5b,c. In general, when researching the properties of copolymers, it is essential to choose a suitable copolymers concentration.

### 3.3. The Properties of Temperature/Reduction Dual-Stimulus Responsive Triblock Polymeric Micelles

#### 3.3.1. Critical Micelle Concentration (CMC) of the Polymeric Micelles 

Because the SPMO triblock copolymers are water-soluble, the copolymers can self-assemble to core–shell micelles in water by the direct dissolution method. In addition, the critical micelle concentration (CMC) is an important parameter for studying the properties of micelles and the stability of drug-loaded micelles. Therefore, we used the fluorescent probe technology with pyrene as the probe and the surface tension method to determine the CMC of SPMO triblock polymeric micelles. Figure 6 shows the CMC of SPMO polymeric micelles with different DP measured by two methods. Figure 6a indicates that the CMC of SPMO1, SPMO2, and SPMO3 measured are 0.005 mg/mL, 0.0075 mg/mL, and 0.0147 mg/mL, respectively, by fluorescent probe technology. On the one hand, it shows that the SPMO polymeric micelles have relatively low CMC, so the micelles can exist stably as a drug carrier for a long time in the human body to achieve the effect of sustained drug release; on the other hand, it is proved that the DP of the hydrophilic segments can affect the CMC of the micelles. In other words, the higher the DP of the hydrophilic segments, the better the water solubility of the copolymers, and then the worse the stability of the micelles, the higher the CMC. Figure 6b shows the CMC of micelles determined by the surface tension method. It can be clearly observed that the measured results are basically consistent with the CMC measured by the fluorescent probe technology, which the low CMC of SPMO polymeric micelles are further confirmed. In short, the dual stimulus-responsive triblock polymeric micelle has the very low CMC, so that it still exists in a core–shell structure micelle state at low concentrations. In conclusion, SPMO polymeric micelles are the qualified carrier for hydrophobic drugs.

#### 3.3.2. Dynamic Light Scattering (DLS) of the Polymeric Micelles

Figure 7 analyzes the reproducibility of the particles of SPMO polymeric micelles by DLS. Figure 7a shows the polymeric particles with different DP are measured at room temperature. It can be seen that the particles of SPMO1, SPMO2, and SPMO3 are 210.2 nm, 190.1 nm, and 164.9 nm, respectively, from the figure. In addition, they are all single-peak curves with narrow distribution, which means that the polymeric micelles formed are uniform in size. Figure 7a shows that the smaller the DP of the hydrophilic segments, the larger the polymeric micelle particles. Because the hydrogen bonds between the hydrophilic chains outside the hydrophobic core and the water molecules are relatively large, the polymeric micelles are so much more stretchable that the hydrophobic cores become larger in internal space. Therefore, the micelle hydrodynamic diameters measured become larger as the DP of the hydrophilic segments decreases. Figure 7b shows the size distributions of SPMO2 polymeric micelles measured under the different stimuli. It can be known from the Figure 7b that the micelle size of SPMO2 under no stimulation, 50 °C, and treatment with 5 mM DTT for 24 h are 190.1 nm, 105.7 nm, and 255 nm, respectively. It can be obviously seen that the particle size of the micelles decreases as the temperature increases. When the temperature is greater than LCST, the hydrophilic segments P(MEO_2_MA-co-OEGMA) gradually transforms into the hydrophobic segments. Therefore, the hydrogen bond with water molecules gradually disappears, causing the shell to shrink and the micelle size to decrease. After treatment with 5 mM DTT, the micelle size increased. Because DTT will reduce the disulfide bond (S-S) inside the copolymers to the sulfhydryl group (-SH), and then the copolymeric structure is destroyed. Therefore, some small molecular aggregates are formed, which increases the size of micelles. Figure 7c is the average particle size change curve of the copolymer aqueous solution at 50 °C and 25 °C heating and cooling cycles. It can be seen from the figure that after multiple heating and cooling cycles that the change in the particle size of the polymeric micelles per cycle is less than 3 nm, showing excellent particle size reproducibility. This phenomenon shows that the temperature sensitivity of thermo-responsiveness of SPMO is dynamically reversible. In summary, Figure 7 shows that the SPMO triblock polymeric micelles have temperature sensitivity and reduction stimulus responsiveness in terms of reproducibility of the micellar particles.

#### 3.3.3. Morphology of the Polymeric Micelles 

Figure 8 shows the morphology of SPMO polymeric micelles observed using transmission electron microscopy (TEM) and their particle size distribution histograms are made, corresponding to the hydrodynamic diameters under three different stimulating environments. According to the TEM image of Figure 8a–c, it can be seen intuitively that the SPMO copolymers self-assembly forming core–shell spherical structure micelles with the P(MEO_2_MA-*co*-OEGMA) as the shell, and the PLLA-SS-PLLA as the core in water. In addition, the diameters measured by Nano Measure 1.2 software are approximately 140 nm, 90 nm, and 200 nm, respectively, under no stimulation, 37 °C, and treatment with 5 mM DTT for 24 h, indicating that the change trends of micelles size are the same after different stimuli. It can be known that the micelles become smaller and more uniformly spherical from the morphology of the micelles in Figure 8b. This phenomenon further confirms that the hydrophilic segment P(MEO_2_MA-*co*-OEGMA) has undergone the change from hydrophilicity to hydrophobicity with the temperature rises, which leads to shrinkage of the shell and reduction of particle size. It can be seen that the micelles become large and uneven non-spherical small molecular aggregates from the morphology of the micelles in Figure 8c. The phenomenon confirms that DTT will destroy the internal structure of micelles and make micelles become the small molecular aggregates with uneven particle size. In summary, dual stimulus-responsive triblock copolymers proved again to be synthesized by observing the morphology of the SPMO copolymers.

Comparing the DLS curve and TEM image of SPMO polymeric micelles, we found that there is the certain deviation in the particle size of the micelles characterized by the two methods in different stimulation environments. The particle sizes measured by DLS are larger than that by TEM obviously. The main reason is that the morphology of polymeric micelles is observed in the dry state by TEM, while the particle sizes of the micelles are measured in the aqueous solution by DLS.

### 3.4. Sol–Gel Transition of the Temperature/Reduction Dual-Stimulus Responsive Triblock Copolymers 

The transformation behavior of sol–gel of SPMO copolymers are studied through vial flip experiment. The sol–gel transition behavior of micelles refers to the process in which the copolymers solution changes from the sol form in a fluid state to the gel form in an elastic non-flowing state, which is also called the gelation behavior of micelles. The minimum temperature at which the copolymeric aqueous solution undergoes transformation is the critical temperature of the micellar sol–gel transformation. Figure 9 is the sol–gel transformation photos taken by the digital camera. Figure 9a shows the sol–gel transition diagram of SPMO copolymers with the mass concentration of 20 wt% but different DP. Through the analysis of Figure 9a, it can be known that SPMO1 and SPMO2 both begin to form gels at 35 °C, and SPMO3 begin to form gels at 40 °C. In addition, all SPMO copolymers form the gels completely at 55 °C. It can be seen that the DP can affect the critical temperature of the sol–gel transformation. In addition, the greater the DP, the higher the temperature of gel formed, and the longer the DP of the hydrophilic segments, the stronger the hydrogen bond with water molecules, which will hinder the three-dimensional network structure physical gel formed by the interaction between hydrophilic and hydrophobic molecules. Figure 9b shows the sol–gel transition diagram of SPMO2 copolymers at different mass concentrations. Figure 9b shows that SPMO2 copolymeric aqueous solutions with mass concentrations of 5 wt%, 10 wt%, 15 wt%, and 20 wt% start to dehydrate in order to form gels at about 45 °C, 45 °C, 40 °C, and 35 °C, respectively. All SPMO copolymers form physical gels completely at 50 °C. It can be seen that the concentration plays an important role in the formation of the gel. In other words, the greater the concentration, the easier the gel is to form. The greater the concentration of the SPMO copolymeric aqueous solutions, the weaker the hydrogen bond between the hydrophilic segment and the water molecules, which can be destroyed at a lower temperature. Therefore, it is easier to form the three-dimensional network structure physical gels by the interaction between hydrophilic and hydrophobic molecules as the crosslinking point. In short, the SPMO dual stimulus-responsive triblock copolymers have the sol–gel transition behavior, and the transition temperature is affected by the DP and concentration.

### 3.5. Drug Release Behavior of the Temperature/Reduction Dual-Stimulus Responsive Triblock Copolymers 

The drug loading and release behavior of SPMO copolymeric micelles are studied using the hydrophobic anticancer drug doxorubicin (DOX) as a model. The drug loading (DL) and encapsulation efficiency (EE) of drug-loaded micelle is obtained using UV–Vis spectroscopy. The DL and EE of the DOX-SPMO2 polymeric micelle are 5.3% and 66.1%, respectively. The particles of DOX-SPMO2 polymeric micelle is 239.2 nm by DLS. Compared with SPMO2 polymeric micelle, the significantly increased diameters of DOX-SPMO2 polymeric micelle indicated that DOX is loaded into the micelle successfully. Figure 10 shows the accumulative release of DOX after the drug-loaded micelles are given different stimulation environments. Figure 10a is the accumulative release amounts of DOX from drug-loaded micelles at different temperatures. It can be seen from the figure that the accumulative release of free DOX reaches 85% within one day; the release of drug-loaded micelles reaches 9.56% within 20 h under no stimulation conditions; the accumulative release of DOX reaches 13.38% within 20 h at 37 °C; the accumulative release of DOX reaches 20.20% within 20 h at 45 °C. Hence, the drug-loaded micelles clearly showed the slow and sustained release behavior compared to free DOX. It shows that the SPMO dual stimulus-responsive triblock copolymeric micelle is the good drug carrier, which can well encapsulate hydrophobic drugs in the hydrophobic core. In addition, the accumulative release of DOX from drug-loaded micelles reached 17.47% within 168 h under no stimulation conditions, the accumulative release of DOX reached 34% within 168 h at 37 °C, and the accumulative release of DOX reached 47.37% within 168 h at 45 °C. It can be found that the drug-loaded micelles show obviously temperature responsive behavior in terms of drug accumulative release by comparison. That is, the higher the temperature, the greater the accumulative release of DOX. The main reason is that when the temperature is higher than LCST, the hydrophilic segment P(MEO_2_MA-*co*-OEGMA) transforms into the hydrophobic segment, which leads to the deformation of the core-shell structure and the reduction of the micelle particle size. Therefore, the DOX in the micellar core are squeezed out, and the accumulative releases of DOX will increase.

Figure 10b shows the accumulative releases of DOX under different concentrations of DTT. It can be known that the accumulative release of DOX reached 32.76% within 168 h under 1 mM DTT. However, the accumulative release of DOX reached 48.31% within 116 h after adding 10 mM DTT, and the accumulative release of DOX are as high as 61.77% within 100 h, when the drug-loaded micelles are given the dual stimulation environment of 37 °C and 10 mM DTT. Comparing the release of free DOX, it can be seen that the drug-loaded micelles in the reducing environment exhibit a slow drug release behavior, which proves that the novel micelle is an excellent drug-loading tool. Comparing the release of DOX under no stimulation conditions, it can be seen that DOX releases faster in the reducing environment. Moreover, the higher the concentration of the reducing agent, the faster the drug release of the drug-loaded micelles, which proves that the novel polymeric micelles can make corresponding stimulus responses to the reducing environment. It also shows that the drug-loaded micelles can be released quickly and largely in tumor cells with the high concentration of GSH in a reducing environment to achieve better therapeutic effects. In general, when drug-loaded micelles are transported in the human body, the micelles exist in the stable core–shell structure due to their low CMC value. Therefore, there is a small amount of drug release. However, when the drug-loaded micelles are transported to the lesion, the core–shell structure of the drug-loaded micelles will change under the stimulation of local high temperature and high concentration of GSH, which forces a great deal of drugs to be released in a short time. Thereby, the precise targeting effect of the drug is achieved, and the purpose of treatment is achieved.

### 3.6. Self-Assembly Behavior of the Temperature/Reduction Dual-Stimulus Responsive Triblock Copolymers and the Drug Release Mechanism of Micelles

Figure 11 is the schematic diagram of the self-assembly behavior of SPMO copolymers in water and the drug release of DOX-loaded micelles under different stimulation environments. When the concentration is lower than the CMC of the micelle, the triblock copolymers P(MEO_2_MA-*co*-OEGMA)-*b*-PLLA-SS-PLLA-*b*-P(MEO_2_MA-*co*-OEGMA) exists in a linear structure, and when the concentration reaches the CMC of the micelle, the linear structure copolymers will self-assemble in water to form the spherical micelles with stable core–shell structure at room temperature. Here the hydrophobic PLLA-SS-PLLA are the core and the hydrophilic P(MEO_2_MA-*co*-OEGMA) are the shell. At this time, the hydrophobic drug DOX are encapsulated in the core, and the hydrophilic segments are in the stretched states because of the hydrogen bond with molecules. When the temperature is greater than the LCST, as the hydrophilic segments change to hydrophobicity, they will shrink and surround the hydrophobic core. Hence, the micellar sizes will decrease, and the DOX are squeezed out from the core. After adding the small molecule reducing agent DTT to the aqueous solution of micelles, the -SS- in the hydrophobic core will be reduced to form -SH, and the micellar structure will be destroyed, so that DOX will be released. In addition, when the temperature is less than the LCST, the hydrophilic segments become hydrophilic again. Hence, the micellar structure will undergo the reversible transformation behavior as the temperature changes. However, since the transition of -SS- and -SH is one-way after adding DTT, the change of micelle structure is irreversible. Because the concentration of GSH in human tumor tissue can reach 10 mM, and the temperature of the diseased tissue is higher than the temperature of the human body, the drug-loaded micelles will release a large amount of drugs quickly under the dual stimulation of the both. This reflects the precise targeting effect of SPMO polymeric micelles. In conclusion, the dual stimulus-responsive triblock polymeric micelles can be used as excellent carriers for hydrophobic drugs.

## 4. Conclusions

In summary, the characterization of ^1^H NMR, FTIR and SEC can prove that had synthesized a temperature/reduction dual stimulus responsive triblock polymer with a specific structure and a certain DP via ROP and ATRP technology. The digital photos recorded the water solubility of the amphiphilic copolymers. The LCST of the copolymers were adjusted to 37 °C. Polymeric micelles have lower CMC indicating that the novel polymeric micelles can exist stably as a drug carrier for a long time in the human body to achieve the effect of sustained drug release. The particle size of micelles under different stimulating environments proved that novel micelles have temperature and reduction responsiveness. The sol–gel transition indicated that the copolymers can undergo the sol-to-gel transition at an appropriate temperature and concentration. Through the study of the accumulative release of DOX under different stimulation environments, it is proved that DOX can be released quickly and largely in the tumor tissue environment. In conclusion, the novel temperature/reduction dual stimulus-responsive triblock polymeric micelle is expected to become the excellent carrier for hydrophobic anticancer drugs, and it has great potential for the controlled drug release and drug delivery.

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
