# Peer review of "Novel Temperature/Reduction Dual-Stimulus Responsive Triblock Copolymer [P(MEO2MA-co- OEGMA)-b-PLLA-SS-PLLA-b-P(MEO2MA-co-OEGMA)] via a Combination of ROP and ATRP: Synthesis, Characterization and Application of Self-Assembled Micelles"

_polymers, 2020, doi:10.3390/polym12112482_

Round 1

Reviewer 1 Report

Title: Synthesis of temperature/reduction dual-stimulus responsive triblock copolymer [P(MEO2MA-co-OMEGA)-b-PLLA-SS-PLLA-b-P(MEO2MA-co-OEGM A)] via ROP and ATRP, characterization and application of micelles (Submitted to Polymers)

   This paper demonstrated the synthesis of dual stimulus responsive triblock copolymer system composed of thermal responsive polymethacrylates having ethylene glycol skeleton and polylactic acid possessing disulfide linkage at mid-chain and the evaluation of drug releasing property of its micelles. Generally, the interpretation of the data draws reasonable conclusion. Overall, this is a well written manuscript, and results presented seem reliable. However, the authors would need to address several important issues before publication. We support the publication on this manuscript with the following major revisions.

  1. Some previous publications on the thermoresponsive polymer of tri(ethylene glycol) methacrylate were missing. I suggest that the authors at least add the following papers as the background: e.g Polymer 190 (2020) 122257, Macromol. Chem. Phys., 218 (2017) 1700048, RSC Adv. 7 (2017) 25199.

  1. Although the authors described “CuBr was synthesized using oxidoreduction of CuSO4 and Na2SO3.”, is it really possible to synthesize without Br salts?

  1. In the footnote of Figure 1, the sample name (SP, SPBr, SPMO) should be used instead of a, b, c.

  1. It is important to show the result of GPC (gel permeation chromatography) curve including the information about the molecular weight distribution as evidence that the object polymer could be synthesized. These results should be provided at each stage.

  1. In the thermoresponsiveness, it is better to show the result of hysteresis including the cooling process behavior.

  1. Concerning about micelle; although an increase in micelle size has been observed after reduction by treatment with DTT, could you observe the similar behavior to utilizing the block copolymer system obtained by complete reduced of disulfide linkage? In addition, what about the stability of micelles in the static state and in the presence of salt?

・The authors should correct some minor errors as below.

* P.2 in Abstract, “disulfifide” → “disulfide”

* P.7 in Table 1, “[SS-DOH]/[PLLA]” → “[LLA]/[SS-DOH] (?)”

* P.7 line 287, “2-Bromo-” → “2-bromo-”

* In Figure 2, there are two “h”s in the upper figure.

* P.13 line 459, the sentence has no period.

Author Response

Dear Reviewer

Thank you very much for your comments and suggestion. We have made a serious revision to the manuscript according to your comments and suggestion, and the revised part is marked in the scarlet letter.

Major corrections:

  1. Some previous publications on the thermoresponsive polymer of tri(ethylene glycol) methacrylate were missing. I suggest that the authors at least add the following papers as the background: e.g Polymer 190 (2020) 122257, Macromol. Chem. Phys., 218 (2017) 1700048, RSC Adv. 7 (2017) 25199.

Reply: Thank you for your comments and suggestion. Literatures about the thermoresponsive polymer of tri (ethylene glycol) methacrylate have been offered and supplied in the introduction section with marked out in the scarlet letter.

  1. Although the authors described “CuBr was synthesized using oxidoreduction of CuSO4 and Na2SO3.”, is it really possible to synthesize without Br salts?

Reply: Thank you for your comments and suggestion. The Br salt is indispensable during the synthesis of CuBr. KBr has been added to the manuscript with marked out in the scarlet letter.

  1. In the footnote of Figure 1, the sample name (SP, SPBr, SPMO) should be used instead of a, b, c.

Reply: Thank you for your comments and suggestion. The sample name (SP, SPBr) have been used instead of a, b, c in the manuscript of Figure 1. The revised part is marked in the scarlet letter.

  1. It is important to show the result of GPC (gel permeation chromatography) curve including the information about the molecular weight distribution as evidence that the object polymer could be synthesized. These results should be provided at each stage.

Reply: Thank you for your comments and suggestion. The data on GPC have been listed in the Table 1. GPC traces of polymer have been shown in the figure 3. Corresponding explanations have been given and marked out in the scarlet letter in the revised manuscript.

  1. In the thermoresponsiveness, it is better to show the result of hysteresis including the cooling process behavior.

Reply: Thank you for your comments and suggestion. The reversibility behavior of polymeric thermoresponsiveness has been shown in Figure 7(c) of the revised manuscript. Its related explanation has also been shown and marked out in the scarlet letter in the revised manuscript.

  1. Concerning about micelle; although an increase in micelle size has been observed after reduction by treatment with DTT, could you observe the similar behavior to utilizing the block copolymer system obtained by complete reduced of disulfide linkage? In addition, what about the stability of micelles in the static state and in the presence of salt?

Reply: Thank you for your comments and suggestion. The answers to several questions raised by the reviewers are as follows.

Thank you for your comments and suggestion. The synthesized copolymers are transparent colloidal substance, which self-assembles to form spherical micelles when dissolved in water. After adding DTT treatment, it can be clearly seen that the spherical structured micelles are destroyed by TEM, which is enough to prove that the SPMO block polymers have the reduction-responsive behavior.

Thank you for your comments and suggestion. The particle size of the previously measured samples are measured again, and it is found that the difference in particle size measured two months ago are less than 3 nm, indicating that novel polymeric micelles are relatively stable under static conditions.

Thank you for your comments and suggestion. On September 30, 2020, SPMO2 micellar solution (c=1 mg/mL) was configured with PBS buffer solution as solvent, and the particle size measured was 187 nm by DLS at room temperature. After 6 days, the particle size measured was 185 nm, indicating that the micelles are relatively stable in the presence of salt.

Minor corrections:

  • P.2 in Abstract, “disulfifide” → “disulfide”

Reply: Thank you for your comments and suggestion. Corresponding error in the manuscript has been corrected and with scarlet letter marked.

  • P.7 in Table 1, “[SS-DOH]/[PLLA]” → “[LLA]/[SS-DOH]”

Reply: Thank you for your comments and suggestion. Corresponding errors in the manuscript have been corrected and with scarlet letter marked.

  • P.7 line 287, “2-Bromo-” → “2-bromo-”

Reply: Thank you for your comments and suggestion. Corresponding errors in the manuscript have been corrected and with scarlet letter marked.

  • In Figure 2, there are two “h”s in the upper figure.

Reply: Thank you for your comments and suggestion. Corresponding errors in the manuscript has been corrected.

  • P.13 line 459, the sentence has no period.

Reply: Thank you for your comments and suggestion. Corresponding errors in the manuscript have been corrected and with scarlet letter marked.

   Kind regards,

Shouxin Liu

Reviewer 2 Report

The subject of the research is not bad, and the research is designed in appropriate way. But the data shown in the article are not properly interpreted or explained. To publish this paper, the author have to re-interpret data.

  1. The title of this article does not show your work well. This title sounds like this article is focused on synthesis, but I think synthesis is not the core part of your work. You need to refine the title.
  2. How about to measure SEC (GPC) for your polymer? There are no evidence that your polymer is block copolymer other than it forms micelles. It might be mixture of homopolymer or homopolymers + block copolymer. Also, if the author shows molecular weight and DP from SEC, it will be much more clear. And also SEC can give us dispersity information.
  3. From Table 1
    1. 2nd column ([SS-DOH]/[PLLA]) values are not understandable. The author wrote there "(n:n)", but in the table, it is just written "20", and it is not clear which block is 20. If the author fix it like 3rd column, it will be better to understand.
  4. From Fig 1 (IR results)
    1. y axis scale of each spectrum is different, and it makes hard to understand the results. eg, black line (a) has too small y axis scale, but red line (b) has too big y axis scale.
    2. the author did not mentioned what are the a, b, and c. 
    3. Arrows in the figures are not clear which band it means.
    4. Some interpret errors found; band from ~3000 cm-1 includes -C-H of CH2. So you have to fix it to "sp3 -C-H stretching" or "-C-H stretching" or "aliphatic -C-H" or so on. 
    5. From line 282, the author says that the peak at 1378 cm-1 of methyl groups indicate the synthesis of homopolymer, but your monomer also has methyl group which gives the peak nearby 1378 cm-1. Please give the IR spectrum of monomer together, so that it will help to understand your explanation.
  5. From Fig 2 (NMR results)
    1. When the author explain the NMR results, some chemical shifts are written with two decimals, but some are one. Please equalize them.
    2. From the figure, proton f and g from PEO will have same chemical shift. The peak f and g from the graph is from the first block of PEO (COO-CH2-) and others (-CH2-CH2-O-).
  6. From Fig 7 (TEM and DLS results)
    1. In DLS results, x axis of (d) and (f) are expressed in Log scale, but not for (e). It will make confusing for the readers. Please use same x axis scales for all the results.
    2. compare to (a) and (b), (b) image has much smaller structures. but corresponding DLS results ((d) and (e)), (e) has much larger particle size. Why?
  7. From Fig 8 (Photographs)
    1. Most of the figures are looks like crushed flat. 
  8. For the reduction stimuli responsive results, it will be better to compare with control sample (P(MMEO2MA-co-OEGMA)-b-PLLA-b-P(MMEO2MA-co-OEGMA), which does not have S-S group).
  9. typo errors
    1. line 103, 115, 258, and 300/ 2,2'dithio-diethanol > 2,2'-dithio diethanol
    2. line 104/ 2,2' -bipyridine > 2,2'-bipyridine
    3. line 115/ Tin(II) > tin(II)
    4. line 147/ methacrylate(MEO > methacrylate (MEO
    5. line 147/ methacrylate(OEGMA > mthacrylate (OEGMA
    6. line 149/ N, N-dimethylformamide > N,N-dimethylformamide
    7. line 261 and 287/ 2-Bromo > 2-bromo
    8. line 275 and 297/ 1HNMR > 1H NMR
    9. line 281/ 1647cm-1 and 1378cm-1 > 1647 cm-1 and 1378 cm-1
    10. need to correct some abbreviation/ FTIR and FT-IR
    11. There are so many typo error, so I didn't try to find the error after these. Please check the typo errors for entire article.

Author Response

Dear Reviewer

Thank you very much for your comments and suggestion. We have made a serious revision to the manuscript according to your comments and suggestion, and the revised part is marked with the blue(or scalet) letter.

  1. The title of this article does not show your work well. This title sounds like this article is focused on synthesis, but I think synthesis is not the core part of your work. You need to refine the title.

Reply: Thank you for your comments and suggestion. Title has been refined in the manuscript with marked out in the blue letter.

  1. How about to measure SEC (GPC) for your polymer? There are no evidence that your polymer is block copolymer other than it forms micelles. It might be mixture of homopolymer or homopolymers + block copolymer. Also, if the author shows molecular weight and DP from SEC, it will be much more clear. And also SEC can give us dispersity information.

Reply: Thank you for your comments and suggestion. The answers to several questions raised by the reviewers are as follows.

  • Thank you for your comments and suggestion. The data on GPC have been listed in the Table 1. GPC traces of polymer have been shown in the figure 3. Corresponding explanations have been given and marked out in the scarlet letter in the revised manuscript.
  • Thank you for your comments and suggestion. On the one hand, the molecular weight of the polymers measured by GPC are basically consistent with the theoretical values, indicating that the block polymers with a specific structure are synthesized instead of a homopolymer or a mixture of homopolymer and block copolymer. On the other hand, their molecular weight distribution (PDI≤1.50) are relatively narrow, which also proves the successful synthesis of SPMO polymers.

  1. From Table 1

(1)2nd column ([SS-DOH]/[PLLA]) values are not understandable. The author wrote there "(n:n)", but in the table, it is just written "20", and it is not clear which block is 20. If the author fix it like 3rd column, it will be better to understand.

Reply: Thank you for your comments and suggestion. Corresponding errors in the manuscript have been corrected and with scarlet letter marked. In addition, "20" represents [LLA]/[SS-DOH] = 20:1.

  1. From Fig 1 (IR results)

(1) y axis scale of each spectrum is different, and it makes hard to understand the results. eg, black line (a) has too small y axis scale, but red line (b) has too big y axis scale.

(2) the author did not mentioned what are the a, b, and c.

(3) Arrows in the figures are not clear which band it means.

(4) Some interpret errors found; band from ~3000 cm-1 includes -C-H of CH2. So you have to fix it to "sp3 -C-H stretching" or "-C-H stretching" or "aliphatic -C-H" or so on.

(5) From line 282, the author says that the peak at 1378 cm-1 of methyl groups indicate the synthesis of homopolymer, but your monomer also has methyl group which gives the peak nearby 1378 cm-1. Please give the IR spectrum of monomer together, so that it will help to understand your explanation.

Reply: Thank you for your comments and suggestion. The answers to several questions raised by the reviewers are as follows.

  • Thank you for your comments and suggestion. The SPBr sample was remeasured by FTIR. The measured results are shown in Figure 1(b).
  • Thank you for your comments and suggestion. The sample name (SP, SPBr) have been used instead of a, b in the manuscript of Figure 1. The revised part is marked in the scarlet letter.
  • Thank you for your comments and suggestion. The peaks of the functional groups indicated by the arrows in Figure 1 have been relabeled.
  • Thank you for your comments and suggestion. The peaks at 2945 cm-1 is the stretching vibration absorption peak of C-H for methyl and methylene, the peaks at 1647cm-1and 1378cm-1 are the in-plane flexural vibration absorption peaks of CH- for methylene and methyl respectively. The peak at 1378 cm-1 is a sign of the presence of methyl groups. They are all marked in Figure 1.
  • Thank you for your comments and suggestion. Because of the functional groups of the temperature-sensitive monomers MEO2MA and OEGMA are the same as the synthesized block polymers, the existence of these FTIR peaks can’t proves that the temperature-sensitive monomers MEO2MA and OEGMA have been successfully copolymerized at random. The successful synthesis of SPMO needs to be verified by 1H NMR spectroscopy (Figure 2).

  1. From Fig 2 (NMR results)

(1) When the author explain the NMR results, some chemical shifts are written with two decimals, but some are one. Please equalize them.

(2) From the figure, proton f and g from PEO will have same chemical shift. The peak f and g from the graph is from the first block of PEO (COO-CH2-) and others (-CH2-CH2-O-).

Reply: Thank you for your comments and suggestion. The answers to several questions raised by the reviewers are as follows.

  • Thank you for your comments and suggestion. Corresponding errors in the manuscript have been corrected and with blue letter marked.
  • Thank you for your comments and suggestion. Proton peak ‘f’ and ‘g’are not same chemical shift. The peaks at 00 ppm(peak “f”) is attributed to the first methylene proton peak of repeating units adjacent to the ester bond of MEO2MA and OEGMA. The peaks at 3.60 ppm (peak “g”) is attributed to the second methylene proton peak of repeating units adjacent to the ester bond of MEO2MA and OEGMA.

  1. From Fig 8 (TEM and DLS results)

(1)In DLS results, x axis of (d) and (f) are expressed in Log scale, but not for (e). It will make confusing for the readers. Please use same x axis scales for all the results.

(2)compare to (a) and (b), (b) image has much smaller structures. but corresponding DLS results ((d) and (e)), (e) has much larger particle size. Why?

Reply: Thank you for your comments and suggestion. The answers to several questions raised by the reviewers are as follows.

  • Thank you for your comments and suggestion. Corresponding x axis of (e) in the manuscript have been expressed in Log scale.
  • Figure 8 (d) and (e) that the micelle size of SPMO2 under no stimulation, 37 °C are 190.1 nm, 137.4 nm, respectively. It indicates that the change trends of micelles size are the same as by TEM at temperature stimuli. The reason why the reviewer are confused about corresponding DLS reaults that the x-axis of the graphs (d, e) are expressed with Log scale.

  1. From Fig 9 (Photographs) Most of the figures are looks like crushed flat.

Reply: Thank you for your comments and suggestion. The author is not proficient in camera technology, which leads to poor results. Hoping the reviewer could understand author sincerely. But, it can been seen that from Figure 9 (a) and (b) the SPMO dual stimulus-responsive triblock copolymers have the Sol-Gel transition behavior, and the transition temperature is affected by the DP and concentration.

  1. For the reduction stimuli responsive results, it will be better to compare with control sample (P(MEO2MA-co-OEGMA)-b-PLLA-b-P(MEO2MA-co-OEGMA), which does not have S-S group).

Reply: Thank you for your comments and suggestion. The P(MEO2MA-co-OEGMA)-b-PLLA-b-P(MEO2MA-co-OEGMA) polymers mentioned by the reviewers are not part of the research content of this work. However, this provides the author with an new idea for the second research work, and the author expresses her gratitude to the reviewers. What’t more, the reduction responsive behavior of SPMO has been demonstrated well in this work.

  1. typo errors
  • line 103, 115, 258, and 300/ 2,2'dithio-diethanol > 2,2'-dithio diethanol

Reply: Thank you for your comments and suggestion. Corresponding error in the manuscript has been corrected and with blue letter marked.

  •  line 104/ 2,2' -bipyridine > 2,2'-bipyridine

Reply: Thank you for your comments and suggestion. Corresponding error in the manuscript has been corrected and with blue letter marked.

  • line 115/ Tin(II) > tin(II)

Reply: Thank you for your comments and suggestion. Corresponding error in the manuscript has been corrected and with blue letter marked.

  • line 147/ methacrylate(MEO > methacrylate (MEO

Reply: Thank you for your comments and suggestion. Corresponding error in the manuscript has been corrected and with blue letter marked.

  • line 147/ methacrylate(OEGMA > mthacrylate (OEGMA

Reply: Thank you for your comments and suggestion. Corresponding error in the manuscript has been corrected and with blue letter marked.

  • line 149/ N, N-dimethylformamide > N,N-dimethylformamide

Reply: Thank you for your comments and suggestion. Corresponding error in the manuscript has been corrected and with blue letter marked.

  • line 261 and 287/ 2-Bromo > 2-bromo

Reply: Thank you for your comments and suggestion. Corresponding errors in the manuscript have been corrected and replaced with its abbreviation. The corrected parts are marked with blue.

  • line 275 and 297/ 1HNMR > 1H NMR

Reply: Thank you for your comments and suggestion. Corresponding error in the manuscript has been corrected and with blue letter marked.

  • line 281/ 1647cm-1 and 1378cm-1 > 1647 cm-1 and 1378 cm-1

Reply: Thank you for your comments and suggestion. Corresponding error in the manuscript has been corrected and with blue letter marked.

  • need to correct some abbreviation/ FTIR and FT-IR

Reply: Thank you for your comments and suggestion. Corresponding error in the manuscript has been corrected and with blue letter marked.

  • There are so many typo error, so I didn't try to find the error after these. Please check the typo errors for entire article.

Reply: Thank you for your comments and suggestion. Author have checked the entire manuscript carfully and corresponding corrected parts are marked with blue.

 Kind regards,

Shouxin Liu

Reviewer 3 Report

In the manuscript entitled “Synthesis of temperature/reduction dual-stimulus responsive triblock copolymer [P(MEO2MA-co-OEGMA)-b-PLLA-SS-PLLA-b-P(MEO2MA-co-OEGMA)] via ROP and ATRP, characterization and application of micelles” by Song et al., the authors have prepared a series of ABA smart amphiphilic triblock copolymers by a combination of ROP and ATRP. Moreover, the molecular characteristics of the obtained copolymers were determined but also the colloidal properties. Finally, these micellar systems were used for the loading and controlled release of a model drug under the combined action of two stimuli.

The idea behind this study is quite interesting, the article is clear and the conclusions are supported by the results. However, some corrections are necessary in order to increase the quality of the paper:

  1. HCl is a hydrophilic drug and not a hydrophobic one thus it is quite difficult to encapsulate this type of drug in polymeric micelles. Please verify if you have used hydrophilic or the hydrophobic form of DOX. Add the detail about the drug in the materials section.
  2. The introduction section is not complete. The authors have cited only one recent article and they must complete this section with several recent articles, such as: https://doi.org/10.1007/s10965-020-02108-2 ; https://doi.org/10.1080/10717544.2020.1726526 ; https://doi.org/10.1080/1061186X.2020.1766474 ;

https://doi.org/10.3390/polym12071450 ;

Moreover, the authors must add some references about another important thermo-sensitive polymer which is poly(N-vinylcaprolactam) (PNVCL), the second most studied after the PNIPAM https://doi.org/10.1016/j.eurpolymj.2019.07.015

  1. The authors must indicate how they have calculated the DP and maybe to provide an equation in order to be clear which peaks they have integrated. Also, the theoretical DP should also be provided in table 1. The molecular characterization of the copolymers is quite scarce…
  2. Line 292-293: the authors claimed that FTIR spectra demonstrate the fact that a random copolymer was obtained. This is not true! Moreover, they have not provided, at least, the theoretical r1 and r2 values for the two co-monomers. If they want to state that their copolymer is random, they must determine these two values for their system.
  3. For biomedical application, the zeta potential must be determined before and after drug loading. Also, the size must be determined after the drug loading.
  4. In section 3.4 the authors use the term “hydrogels” in order to describe their system but this is not true. As no crosslinking was carried out, they are not hydrogels but only physical gels. The term hydrogels must be deleted.
  5. Section 3.5: the authors must calculate two essential information about a drug-loaded system, such as: drug encapsulation efficiency (DEE) and drug loading capacity (DLC). Moreover, a FTIR spectrum can indicate the efficient loading of drug and if there are some interactions between the drug and the copolymer sequences which might have an influence on the drug release rate.

Minor corrections:

  1. Line 18: delete “which”
  2. Line 23: replace digital photos by visual observation
  3. Line 24: spectroscopy, which proved that the ….
  4. Line 26: micelles were
  5. Replace the term “what’s more” with “in addition” in all the manuscript
  6. Line 39-40: the polymeric micelles are not all biocompatible; they have attracted attention in the biomedical field due to their small size and loading capacity of hydrophobic drugs. Please correct the sentence.
  7. Line 62: doxorubicin
  8. Do not use “we” in a scientific article. Replace “as we all know” with “it is well know that” in all the manuscript
  9. Line 102: who’s alladin?!
  10. Line 191: the CMC using…
  11. Line 192: were configured, and the concentration of pyrene…
  12. Line 197: the CMC using…
  13. Complete the caption of fig 1, 2
  14. Indicate the concentration for the curves given in fig 4 a
  15. Line 363: replace “form” with “to”
  16. Line 376: replace “are measured” by “determined”
  17. Line 377: replace “known” with “observed”
  18. Line 392-396: replace “hydrodynamic diameters” with “size distributions”
  19. Line 415: self-assembly forming..
  20. Please use log scale for the figure 7 e
  21. Line 545: the characterization of copolymers by NMR, …
  22. Line 548-549: polymeric micelles have low CMC values indicating that…
  23. Line 558: these micellar systems cannot be used for precise targeting as they are not ligand-functionalized, instead they have a controlled drug release. Please correct the sentence.

In view of the above, I recommend the publication of this manuscript only after major revisions.

Round 2

Reviewer 1 Report

Authors have improved the manuscript according to my previous recommendation, and I am in favor of publication.

Author Response

Dear Reviewer Thank you very much for your trust. Kind regards, Shouxin Liu

Reviewer 2 Report

  1. Using the terms "Dispersity (ÐM)" and "Size exclusive chromatography (SEC)" are recommended instead of PDI and GPC by IUPAC. please refer following or IUPAC web site; https://doi.org/10.1351/PAC-REC-08-05-02
  2. Again for NMR (Figure 2), I just confused to understand the structure you drawn. I'd like you to recommend to draw the structure like -COO-CH2CH2-(OCH2CH2)8-OCH3, and mark 'f' for the first CH2 from right after ester, and it may better to understand.
  3. Few more typo error found
    1. line 20, 108, 120, 305, and 308; 2,2'dithio-diethanol > 2,2'-dithio diethanol
    2. line288 and 295; FT-IR > FTIR
    3. line 403, 413; Figure 6 (a), Figure 6 (b) should be bolded.
    4. And some space missing;
      1. line239; Doxorubicin(DOX) > Doxorubicine (DOX)
      2. line 241; DMF (4 mL) ,DOX> DMF (4 mL), DOX
      3. line 375; 1mg/mL > 1 mg/mL
      4. line 422; 50℃, 25℃ > 50 ℃, 25 ℃
      5. line 454; stimulation(a,d) > stimulation (a,d)

Reviewer 3 Report

Q1: I don’t understand why the authors have not used directly the hydrophobic form of Dox and they got complicated instead with Dox-HCl and TEA…However, in the experimental section this explanation must be inserted.

Q2: please revise ref 8 and 17
